# Chemerin Forms: Their Generation and Activity

**DOI:** 10.3390/biomedicines10082018

**Published:** 2022-08-19

**Authors:** Lei Zhao, Lawrence L. Leung, John Morser

**Affiliations:** 1Division of Hematology, Stanford University School of Medicine, Stanford, CA 94305, USA; 2Veterans Affairs Palo Alto Health Care System, 3801 Miranda Avenue, Palo Alto, CA 94304, USA

**Keywords:** chemerin, proteases, obesity, diabetes

## Abstract

Chemerin is the product of the *RARRES2* gene which is secreted as a precursor of 143 amino acids. That precursor is inactive, but proteases from the coagulation and fibrinolytic cascades, as well as from inflammatory reactions, process the C-terminus of chemerin to first activate it and then subsequently inactivate it. Chemerin can signal via two G protein-coupled receptors, chem1 and chem2, as well as be bound to a third non-signaling receptor, CCRL2. Chemerin is produced by the liver and secreted into the circulation as a precursor, but it is also expressed in some tissues where it can be activated locally. This review discusses the specific tissue expression of the components of the chemerin system, and the role of different proteases in regulating the activation and inactivation of chemerin. Methods of identifying and determining the levels of different chemerin forms in both mass and activity assays are reviewed. The levels of chemerin in circulation are correlated with certain disease conditions, such as patients with obesity or diabetes, leading to the possibility of using chemerin as a biomarker.

## 1. Introduction

This review will provide a brief account of the basic biology of chemerin that has been delineated since its discovery two decades ago, before considering the importance of the different forms of chemerin and the issues involved in obtaining accurate determinations of them. It will point to some outstanding areas where progress is needed and note some pitfalls in the acquisition of samples for chemerin determinations, along with some of the problems with frequently used techniques for the measurement of the different forms. At the time of writing, PubMed identifies >1200 papers using chemerin or rarres2 as the search term that have been published in the two decades since its original discovery. This is not going to be an exhaustive review of the chemerin literature in its entirety but will focus instead on a few topics of general interest.

In this review, a chemerin form is defined as the full-length secreted protein and its proteolytic cleavage products, and is preferred to chemerin isoforms, as an isoform is often employed to imply alternative gene products resulting from duplicated genes, or alternative splicing leading to different products. In the case of chemerin, the different forms all derive from the proteolytic cleavage(s) of the secreted full-length protein. The chemerin protein forms are referred to by their C-terminal amino acid number followed by the single letter amino acid code.

## 2. Chemerin System of Genes, Proteins, and Receptors

In a study identifying genes by subtractive hybridization in skin graft cultures that responded to treatment with a synthetic retinoid, tazarotene, a novel mRNA was identified that was named *TAZAROTENE-INDUCED GENE 2* (TIG-2) [1]. This study did not identify the role of the encoded protein, but to our knowledge this is the earliest report of the chemerin protein and gene.

An orphan receptor now known as chemerin receptor 1 (chem1, also known as chemokine-like receptor 1, CMKLR1; chemR23; DEZ) was identified as a receptor for TIG-2 protein, renamed as chemerin (also known by its gene name, retinoic acid receptor response protein 2, *RARRES2*) in studies in which biological fluids were evaluated for activity on cells overexpressing chem1, and those with high activity were purified through a multi-step procedure to yield a single protein. When that purified protein was analyzed by tryptic digestion, the sequence was identical to the protein encoded by the *RARRES2* gene (TIG2) except for missing a few amino acids at the C-terminus [2,3,4].

The sequence of chemerin consists of a signal peptide allowing secretion, a core with 3 disulfide bonds, and N- and C-terminal tails (Figure 1a). No 3D crystal structure has yet been reported, although a preliminary NMR structure with most residues assigned but lacking the C-terminus is available [5]. A model is available in alphafold, predicting that there is a central domain with a relatively rigid structure containing the three disulfide bonds, while the N- and C-termini do not have tight rigid features (Figure 1b). This central domain has some structural similarities with the cathelicidin family of anti-microbial peptides as well as with the cystatin family of cysteine protease inhibitors [2,6]. It was clear from the first reports describing chemerin that the full-length protein needed to be proteolytically cleaved within the C-terminal tail to generate activity [2,4,7], and forms that ended at serine 157 were suggested to be the most active on the chemerin receptor, chem1. Forms that have been identified either from in vivo biopsy samples or from in vitro enzymatic studies are shown in Figure 1a.

Upon binding chemerin, chem1 signals via the Gi/o family of G proteins, thereby inhibiting production of cyclic AMP while increasing production of IP3 and calcium mobilization, in addition to the activation of phospholipase C and the PI3 kinase and MAPK pathways [8]. Stimulation of chem1 can also activate the RhoA/ROCK pathway [9] and induce binding of arrestins to chem1 [10,11], which can both terminate GPCR signaling and function as a framework for assembling signaling complexes [12].

Chemerin has been linked to two additional receptors, chemerin receptor 2 (chem2, also known as GPR1) and chemokine receptor-like 2 (CCRL2). The nomenclature of chemerin receptors used here is that determined by the International Union of Basic and Clinical Pharmacology Committee on Receptor Nomenclature and Drug Classification based on the receptor’s cognate ligand [13]. Both chem1 and chem2 are members of subclass A8 belonging in the rhodopsin class of G protein-coupled receptors (GPCRs). CCRL2 does not encode a full serpentine G protein-coupled receptor GPCR and is not a signaling receptor but binds chemerin and can present it to the signaling receptors chem1 and chem2 [14]. The three receptors and chemerin together comprise the chemerin system.

## 3. Chemerin System Evolution

The chemerin gene (*RARRES2*) is found in all tetrapods that have been sequenced but there is no gene annotation for chemerin in fish (Figure 2). The gene consists of 6 exons of which 5 encode the protein sequence. The pattern of introns and exons as well as the encoded protein sequences have been conserved suggesting that chemerin has a significant role in tetrapod biology. In particular the C-terminal tail, responsible for binding to its receptors, chem1 and chem2, has been conserved. The gene has four exons, with the first exon encoding the signal sequence and most of the N-terminal tail, the central core being encoded by the other three exons, with the last exon encoding the final part of the central core and the C-terminal tail. In humans, the *RARRES2* locus is on chromosome 7 on the antisense strand and on chromosome 6 in mice.

In both humans and mice there is a coding gene on the sense strand, leucine rich repeat containing 61 (LRRC61) with it 3′ terminus close to the 3′ terminus of chemerin. In humans but not mice there is also an lncRNA present in the middle of the chemerin locus on the sense strand as well as a pseudogene, ZBED6CL that overlaps with the LRRC61 gene (Figure 2a). In both humans and mice five alternative spliced chemerin mRNAs have been described, but in both species there is one major coding form.

Chem1 and chem2 are similar to each other and belong to the same clade within the GPCR family, clade A8, which includes receptors for small peptides such as bradykinin and apelin. Both chem1 and chem2 genes can be identified not just in tetrapods but also in bony fish, and chem1 can be found additionally in cartilaginous fish. In addition to chemerin, chem1 and chem2 have alternative ligands for these receptors such as the resolvins E1 and E2 [17,18,19], and they may be a clearance receptor for amyloid beta in Alzheimer’s disease [20].

The chemerin system has been studied in model organisms such as mice and rats, as well as humans. In addition, there are data on the chemerin system in cattle, pigs, and chickens [21].

## 4. Chemerin mRNA Regulation

Although the chemerin gene was originally identified as being responsive to a retinoic acid receptor ligand, tazarotene [1], there have been few follow-up studies on the chemerin promoter and control of chemerin mRNA expression. The major regulatory elements controlling constitutive expression were shown to lie in the region between −252 and +258 bp around the start of mRNA transcription at +1 bp [22]. Differences in responses to IL-1b and oncostatin-M were found between hepatocytes and white adipocytes in which there was increased transcription at the chemerin promoter in adipocytes but not in hepatocytes. Those changes correlated with changes in methylation, and the list of transcription factors predicted to interact with the chemerin promoter also differed between the cell types.

## 5. Different Forms of the Chemerin Protein

The human chemerin cDNA encodes an open reading frame of 163 amino acids translated into preprochemerin (Figure 1 and Figure 3a), of which the first 20 comprise the signal peptide which is removed upon secretion to generate the 143 amino acid protein terminating at Ser163 [23]. When it was isolated from human ascitic fluid secondary to ovarian or liver cancer, the analysis of chemerin protein by tryptic digestion followed by mass spectrometry showed that the protein’s C-terminus was at Ser157. This was different from the C-terminus predicted from its encoding cDNA, which extends a further 6 amino acids to Ser163 [2]. Purified chemerin proteins from hemofiltrate were also found to be truncated compared to the full-length protein and were identified as terminating at Phe154 [3].

In subsequent studies on serum and plasma, the major form present was found to be the precursor, chem163S. The liver produces and secretes the precursor into the circulation, shown by the treatment of rats with an anti-sense oligonucleotide that blocks chemerin synthesis specifically in the liver. The anti-sense treatment resulted in circulating levels of chem163S being substantially reduced on Western blots [24]. 

All forms of chemerin bind to glycosaminoglycans, such as heparin, which has been used as a one-step affinity purification to clean up samples [25]. Therefore, it would be expected that there would be a pool of chemerin bound to the glycosaminoglycans in the extracellular matrix in tissues and on blood vessels. There are no data available on this possibility nor on the pharmacokinetics of chemerin, even in reports in which mice have been treated with chemerin. 

Subsequently, a variety of chemerin forms have been produced by either direct expression of them or proteolytic cleavage of a purified longer form, allowing comparative studies of the different forms. These studies have described the activity of the different forms in detail and have clarified the relative activities of the different forms on chem1 using either chemotaxis or calcium transients as the readout [26,27]. 

Unfortunately, similar detailed studies defining the activities of different chemerin forms are not available for chem2, although the signaling of chem1, chem2, and CCRL2 has been compared [9,10]. The results suggest that chem157S binds to chem1 with a K_D_ of 0.88 nM and to chem2 with a K_D_ of 0.21 nM, but signaling through chem2 may be more biased towards β-arrestin pathways than G protein pathways while the Rho/ROCK and MAPK/ERK pathways may be equivalently activated by the two receptors [28]. Although dimerization of GPCRs into homo- and hetero-dimers and maybe higher oligomers can occur, the possibility of dimerization of chem1 and chem2 into either homo- or hetero-dimers and any consequent effects on signaling have not been investigated [29,30].

A nonapeptide equivalent to amino acids 148–157 (sequence in red in Figure 3a and depicted in Figure 3b) is a very weak competitor of the binding of chem157S to chem1 or chem2 but is an effective competitor for itself. In contrast, this nonapeptide possessed good activity at chem1, almost equivalent to that of the protein with a C-terminus at amino acid Ser157 (chem157S) [26]. Carefully characterized chem157S prepared from mammalian cells gave an EC_50_ of low nM on chem1, with the difference in activity between the nonapeptide and chem157S being ~3-fold [26,27]. In assays in which the G protein signaling pathways were analyzed, this nonapeptide had greatly reduced activity compared to chem157S.

This data suggests that most, if not all, of the binding energy between chemerin and chem1 was located in that C-terminal tail terminating at Ser157. Further truncations of the peptides resulted in a loss of binding and activity, highlighting the importance of the C-terminal serine at position 157 [26]. Also important is Phe156, the penultimate residue in the active nonapeptide, based on the activity of an alanine substitution at that position as well as removal of the C-terminal Ser157.

Accurate determination of binding constants and activity values such as K_D_ and EC_50_ for different chemerin forms or peptides requires the preparation of several independent lots of the protein or peptide, followed by an analysis of purity by methods allowing demonstration that the samples are a single molecular entity comprising a monomolecular solution of known and high purity and concentration. These independent lots can then be assayed multiple times in order to accurately estimate the K_D_ and EC_50_. Most preparations of chemerin proteins have not been analyzed by methods that would allow detection of aggregates, such as non-reducing SDS-PAGE or size exclusion chromatography. These analyses are especially critical for *E. coli*-produced material as chemerin is commonly produced as an inclusion body requiring solubilization and refolding. Evaluation of the integrity and purity of chemerin proteins and peptides is complicated by the lack of activity assays that evaluate the properties of the whole molecule, as chem1 and chem2 both respond to the nonapeptide. This means that activity on those receptors is not indicative of the whole protein being in the correct conformation.

The solubility of peptides derived from C-terminus of chemerin may also be an issue. The sequence in chemerin between amino acids #148–156 (FYFPGQFAF) is hydrophobic, and peptides, including the nonapeptide, containing them may present solubility issues in aqueous solutions, thereby preventing accurate determinations of binding and activity parameters.

A 20-mer peptide whose C-terminus is identical to the nonapeptide has also been demonstrated to be active and can inhibit steroidogenesis in both rat testes and ovaries [31].

A defined structure of the nonapeptide has been proposed that consists of anti-parallel strands linked with a turn. This allows the formation of a hydrophobic core and may stabilize the amino acids that interact with the receptors. A model of the interaction of this peptide with chem1 has been developed based on mutational studies of chem1 and varying the composition of the peptide, showing that the critical binding is between F^8^ in the peptide (equivalent to residue 156F in chemerin) and a hydrophobic pocket in chem1, and F^6^ (equivalent to residue 154F in chemerin) with Y268 in chem1 [32]. The evidence for the model of the peptide was strengthened by a synthesis of cyclic peptides based on the model which retain activity. Those cyclic peptidomimetics can be modified to be stable in plasma for more than 48 h by modifying the N-terminus by substituting L-Tyr with D-Tyr [33].

The nonapeptide has also been used as the basis for designing an imaging agent for PET scanning by attaching a chelating agent via a linker to the N-terminus of the peptidomimetic [34]. In this case the peptidomimetic has been modified extensively with unnatural amino acids to prevent degradation while retaining activity. The resulting agent could be used to image DU4475 cancer cells on the flank of a mouse while not producing a clear image of A549 cells on the opposite flank [34]. The fact that DU4475 cells express chem1 while A549 cells do not suggests the specificity of the binding of the labeled peptidomimetic. The peptidomimetic without the chelator and linker has been used to demonstrate that it stimulates the growth of human colorectal cancer cells in a mouse xenograft model [35]. 

The key role of the C-terminal peptide that terminates at Ser157 has been confirmed in studies of the activity of different chemerin forms using the full-length protein on human chem1, which showed that chem157S was most active [27,36]. In contrast, chem158K and chem156F both possess about 5% of chem157S while the precursor, chem163S, has <5%, and chem155A is inactive. In a signal transduction assay, however, using COS cells transiently transfected with chem1, chem156F was found to be equipotent to chem157S [37]. Studies of the activity of different chemerin forms on chem2 have not been carried out in similar detail. 

These studies require highly purified proteins because chem157S is so much more active than the other forms on chem1. If the sample of a purified form other than chem157S contains any trace amounts of chem157S, whether generated during production, purification, or in the assay, the activity of the sample will be overestimated. The same points also apply to the nonapeptide as they can also lead to overvaluing the activity of that form. In addition, it is important that the assay does not process the material being tested into an active form, so the reliability of any assay for this purpose is increased if the assay is fast, such as the measurement of calcium transients.

The sequence of the active nonapeptide is completely conserved between mouse and human, while there is overall high homology throughout the proteins (Figure 3a). It is notable, however, that the relative activities of mouse chem156S and chem155F differ from that of the equivalent human chem157S and chem 156F on either human or mouse chem1. Both mouse chem156S and human chem157S are equally active on chem1 from either species, but mouse chem155F is equipotent to mouse chem156S while human chem156F possesses only ~5% of the activity of human chem157S [9,38]. This suggests that in mice, chem155F may play a more important role as a ligand for chem1 than chem156F does in humans.

The nonapeptide has been used successfully in animal models to demonstrate differences from untreated animals in models such as abdominal aortic aneurysm and atherogenesis [39,40]. The data from the use of both the unmodified nonapeptide as well as the modified analogs support the concept that the key interactions are between F156 and/or S157 and chem1 and chem2.

Most papers reporting chemerin activity have been centered on chem1, although more recent reports have provided similar data sets for chem2. Chemerin is secreted as a precursor (chem163S) with low activity, but chem157S is fully active, and chem156F has partial activity. Chem163S does not spontaneously generate chem157S, which implies proteolytic processing of chem163S into its active forms and then their subsequent inactivation [7,26,27]. Many enzymes from the coagulation, fibrinolytic, and inflammatory systems have been identified that will process chem163S into active forms as well as inactivate those forms [7,41,42].

While the C-terminal tail of chemerin can be of different lengths, the N-terminal amino acid is E20 with no variation reported, as determined either by tryptic digest followed by MALDI-TOF or by Edman degradation [7,27]. This suggests that the N-terminus is relatively proteolytically stable.

A peptide (C15) was identified as being able to inhibit macrophage migration in a screen of 15–20 amino acid peptides based on high performance throughout assay. This peptide was from the C-terminus of chemerin containing residues A141-A155. It was very potent in inhibiting macrophage migration with a U-shaped dose response curve and maximum effect between 0.1–1 nM compared with full length chemerin with a maximum of 1–10 nM [43]. Treatment with this peptide, C15, was able to ameliorate the damage in a murine model of ischemic/reperfusion injury [44,45]. However, the inhibitory effects of chemerin on macrophages ex vivo could not be repeated by an independent group even though those macrophages express chem1 [46].

Completely separate to chemerin’s signaling via the processed C-terminal tail binding to its cognate receptors is its activity as an anti-bacterial defensin [47]. Unlike the interactions with chem1 and chem2, the anti-bacterial properties of chemerin do not depend on cleavage of the C-terminus. Instead, the activity is located within the disulfide bonded core of chemerin in the sequence from V66 to P85. This peptide alone is sufficient for strong inhibition of *E. coli* growth.

## 6. Chemerin Expression and Activity in Cells and Tissues

Chemerin was first identified by its ability to trigger chem1 signaling and that property is still regarded as its canonical characteristic, although its activity as a defensin (see below) and on chem2 ought to be included among the intrinsic roles of chemerin. The interactions of chemerin with its signaling receptors, chem1 and chem2, depends on proteolytic cleavage of chemerin’s C-terminal tail [7] discussed in detail above. The assays originally used in the studies of chemerin purification [2,3,4] were to add samples to cells over-expressing chem1 and utilized signals from either a transfected reporter gene such as apoaequorin [48] or from endogenous signals such as calcium transients. These data suggested a function for chemerin as an immunomodulator, because chem1 is found in some classes of dendritic cells and other immune cells [2,4,49]. Subsequently the number of cell and tissue types shown to be responsive to chemerin has been expanded to include, for example, adipocytes, fibroblast-like synoviocytes, and vascular smooth muscle cells [50,51,52]. There is now well-substantiated evidence of the importance of chemerin in many more physiological processes than inflammation. For instance, chemerin plays a role in white and brown adipocyte development and function [50,53,54], in blood pressure control in rats [51,55], in skin keratinocytes [56,57], and in control of steroidogenesis in both males and females [31]. In the last of these examples, it has been shown that the relevant chemerin receptor is chem2 and not chem1 [58,59].

In humans, chemerin is also expressed in several tissues, but its expression is dominated by the liver, with its high levels of mRNA and being the largest organ in the body (Figure 4a). Other tissues with high levels of chemerin mRNA expression include the adrenals (endocrine tissue) and pancreas with adrenal glands possessing 2400 normalized transcripts per million (nTPM) compared to 2000 nTPM in the liver. 

In blood, most of the circulating chemerin is produced by the liver, but the majority of chemerin found in circulation is the inactive precursor chem163S [25,61,62]. Activation of that chemerin can occur during coagulation, fibrinolysis, or inflammation, as the enzymes responsible for cleaving chem163S are themselves activated from their inactive precursors [7,42]. When coagulation and inflammation are happening as local processes, thereby providing host defense at sites where bleeding or contact with a pathogen or other toxic agent is ongoing, the active chemerin forms will also be expected to have effects locally. These effects include those on dendritic or other immune cells, modulating immune response, or on vascular smooth muscle cells causing vasoconstriction.

In contrast to the systemic production of prochemerin by the liver, there is also local production of prochemerin in some tissues. In those tissues where prochemerin is locally produced, it may be activated and subsequently interact with its receptors to regulate the properties of that tissue. This has been demonstrated in the perivascular arterial fat by using anti-sense oligos to reduce chemerin mRNA expression, either in the whole body or specifically in the liver [55]. The findings were that the chemerin regulation of vasoconstriction was totally dependent on local prochemerin production, and the prochemerin in the blood produced by the liver did not affect the vasoconstriction. An independent study in rats showed that plasma levels of chemerin did not correlate with blood pressure, emphasizing that the plasma pool of chemerin is unlikely to be critical for this function of chemerin [63].

Chemerin is expressed in both the tanycytes and the ependymal cells of the hypothalamus, and these cells express both the thyroid hormone and retinoic acid affecting the photoperiodic response. Photoperiodicity also affects chemerin expression in the rat hypothalamus where long days increase chemerin expression compared to short days, suggesting that this could influence energy balance and body weight [64]. This has been confirmed in rat experiments in which a shRNA that reduces chem1 mRNA was injected into the arcuate nucleus of the hypothalamus in rats, resulting in acute weight loss and the chronic inhibition of weight gain [65]. These data suggest that chemerin via chem1 exerts its effects on energy homeostasis via central regulation as well as in the periphery [66,67].

The chemerin system is likely to be dependent on local production and activation in tissues including brain, skin, white and brown fat, the synovium, and the peritoneum, with the blood pool of prochemerin only playing a major role if there is a breakdown in the endothelial barrier function.

The tissue distribution of the chemerin receptors provides clues into which tissues will be responsive to chemerin. Tissues producing high levels of chemerin such as the liver do not necessarily have high levels of either chem1 or chem2, suggesting that these organs are secreting chemerin for it to be active in other organs which do not have endogenous chemerin production. Conversely, there are tissues with high levels of chem1 or chem2 expression with undetectable chemerin expression. The choroid plexus in the brain possesses high levels of chem1 and undetectable chemerin production, while the kidney has high levels of chem1 expression with low levels of chemerin expression, suggesting that these tissues are sensitive to chemerin produced elsewhere or produced locally when there is a need for a host defense reaction.

Chem1 has been shown to be expressed in many tissue types by mRNA expression studies, exemplified by the data in the human protein atlas generated by immunohistochemistry (Figure 4c). Chem2 mRNA has a different tissue distribution to chem1 (Figure 4d), but unfortunately without protein data being available. It is, however, expressed in many tissues at least at the mRNA level. CCRL2 is also expressed at both the mRNA (Figure 4b) and protein level in multiple tissues. 

## 7. Proteolytic Enzymes in Chemerin Processing

Early studies implicated many proteases in the physiological processing of chemerin including enzymes from the coagulation cascade, fibrinolytic, and inflammatory systems, as well as others such as the angiotensin converting enzyme (ACE) [7,68]. More detailed studies have led to the development of cleavage schemes starting with prochemerin and eventually resulting in inactive degraded chemerin forms (Figure 5) [69]. Compared with other protease substrates with very tight specificity resulting in cleavage by only a single enzyme, either physiologically or pathophysiologically, the chemerin C-terminal tail can be cleaved by a variety of proteases.

Chemerin is highly conserved throughout its sequence from N-terminal E20 to residue 157 (human numbering), but the remaining half-dozen residues have much lower homology. This may be because the constraints on variations in this sequence are lower than elsewhere in chemerin, as the only function of this small peptide may be to provide accessible protease sites, and ensure that the cleaved fragment should not act as an inhibitor of protease after its cleavage. The role of this small peptide may be similar to the fibrinopeptides as proteolytic targets. Fibrinopeptides also have much higher rates of evolution than the remainder of the fibrinogen molecules [70,71].

In plasma (Figure 5), Factor XIa is the most likely enzyme to carry out the first processing step of converting chem163S to chem158K in blood, but plasmin can also catalyze this reaction. One of the two basic carboxypeptidases, carboxypeptidase B2 (CPB2) and carboxypeptidase N (CPN), present in plasma, removes the C-terminal lysine to generate fully active chem157S [41,42]. CPN is constitutively active while CPB2 (also known as thrombin-activatable fibrinolysis inhibitor TAFI) is activated by the thrombin/thrombomodulin complex as well as by thrombin alone, and plasmin bound to glycosaminoglycans [72,73]. Under the conditions in which chemerin is being activated by FXIa, CPB2 will also be activated [74]. The enzymes responsible for the degradation of chem157S have not yet been fully delineated but might include chymase.

An intermediate in the formation of chem158K was identified in these studies, chem162R, which can be generated by either Factor XIa or thrombin [42]. Chem162R is inactive, similar to prochemerin, and is processed to chem158K by Factor XIa but not thrombin. It is not an obligate intermediate in the generation of chem158K. Chem162R is homologous to the commonest circulating form of mouse prochemerin, mouse chem161K [38], but there have been reports of full length mouse prochemerin in samples from adipocyte-conditioned media [75].

In tissues, the enzymes processing chemerin may be different from those in blood. In adipocytes, elastase and tryptase are involved in the processing of chemerin [76], while in skin, kallikrein 7 cleaves chem163S [37]. In synovial fluid of rheumatoid arthritis patients, chem156F was the dominant active form and was produced by chymase cleavage [36]. This suggests that the enzymes cleaving chemerin to generate activity will depend on the tissue, resulting in either chem157S or chem156F as the active form.

Active chem157S can also be produced directly from chem163S by cleavage with cathepsins K and L or human leukocyte elastase, with an intermediate such as chem158K [77]. Similarly, chem163S can be proteolyzed to generate chem156F by cathepsin G, chymase, or kallikrein 7 [36,37,78]. There are no reports of enzymes that convert chem157S into chem156F or either of those active forms into inactive chem155A. In experiments on a chemerin decapeptide representing P145–F154, it was shown that ACE’s dipeptidase activity can process the decapeptide into smaller fragments [68]. Other enzymes such as chymase, elastase, and cathepsins K and L will degrade chemerin forms longer than chem156F to smaller degraded forms [7,78].

This description of the activity of various proteolytic enzymes on different chemerin forms does not address the conditions under which that reaction might be physiologically relevant. For some studies, enzyme kinetic data are available, including the generation of chem156F from chem163S by chymase [36], or by factor Xia, in both cases showing that the enzymes’ cleavage of chemerin proceeded at a similar rate to cleavage of their canonical substrates [36,42]. Under the conditions of using a 100:1 substrate to enzyme ratio incubated at 37 °C for 30 min followed by MALDI-TOF and SDS-PAGE analysis, from a panel of 21 proteases, only factor XIa, plasmin, kallikrein, and tryptase were found to cleave chem163S to generate chem158K [42]. In addition, elastase, MMP8, protease 3, and tryptase cleavage analyzed on SDS-PAGE produced bands of degraded chemerin smaller than chem163S. The subsequent removal of the C-terminal residue of chem158K by CPB2 or CPN has also been shown to occur at a physiologically relevant rate [41]. Moreover, elastase, MMP8, protease 3, and tryptase cleavage produced smaller bands of degraded chemerin. These data suggest that although cleavage of the C-terminal tail of chemerin can be detected following incubation with a variety of enzymes under in vitro conditions, the number of enzymes capable of activating chem163S is limited, and the specific enzyme will depend on the tissue.

Some studies have extended the in vitro studies above by investigating the proteases involved in chemerin processing in cells and tissues. In the in vitro experiments, single purified enzymes were tested in the absence of other enzymes and without any potential cofactors or protease inhibitors, but these will be present in assays studying cells and tissues. 

In adipocytes, where active chemerin is essential for differentiation, tryptase and neutrophil elastase, and possibly aminopeptidase, are involved in generating active chemerin [75]. Stimulation of the cells with TNF-α increases the levels of elastase and tryptase. 

In skin, kallikrein 7 is responsible for the formation of active chem156F with its inhibitor, vaspin (serpinA12), abrogating its formation [37]. In normal skin, the cells producing prochemerin were located in the basal layer and kallikrein 7 was found in stratum corneum, but in skin biopsies from patients with psoriasis, both prochemerin and kallikrein 7 were co-expressed in the complete epidermis. 

In synovial fluid from patients with rheumatoid arthritis, the principal chemerin forms identified were chem158K and chem156F [25,36]. Chymase is the key enzyme in producing chem156F, either directly from prochemerin or via chem158K as an intermediate.

There are several other chemerin forms such as chem155A, chem154F, chem152G, chem144D, and chem125R that have been identified in biopsy samples from blood, adipose tissue, or other tissues, but the enzymes involved in their physiological or pathophysiological formation are still unknown. In vitro experiments have found enzymes that can generate chem155A, chem154F, and chem152G, but so far enzymes that can lead to the formation of chem144D and chem125R remain unknown.

Most of the enzymes involved in processing chemerin are themselves produced as inactive precursors that need to be activated. In blood, the activation of the enzymes in the coagulation and fibrinolytic is well described, but in the other tissues it has not been demonstrated yet how the precursors are activated. In adipocytes and other tissues, one possibility that has been investigated is that the activator enzyme might be a dipeptidyl peptidase IV (DPP-IV), but experiments with inhibitors showed that DPP-IV was not key [76]. 

Based on modeling and mutational studies, ionic interactions occur in prochemerin between the basic residues (K158 and R162) in the C-terminal tail and a glutamate patch in an adjacent helix, stabilizing the structure of the C-terminal tail [37]. When prochemerin is activated by the removal of the C-terminal tail, the new C-termini are able to interact with the chemerin receptors. This model is not supported by the model in alphafold (Figure 1b). 

## 8. Methods for Determining Levels of Chemerin Forms

The early reports of the characterization of chemerin described it as possessing different and mixed C-terminal amino acids, although the N-terminus was always E20 [2,3,4]. It was unclear from these reports if the different C-termini were due to the different source material or due to ex vivo processing. Subsequent studies have highlighted the lability of chemerin’s C-terminal tail to proteases and underlined that ex vivo processing can occur. In addition, different tissue compartments may have different chemerin forms present due to the diverse proteases that might be present. Given this susceptibility of chemerin to proteolysis, it is critical to ensure that no ex vivo processing occurs between sample acquisition and assay. This can be done by including protease inhibitors from the moment at which the biopsy is acquired, and handling the samples in a manner that does not allow any protease activity to occur.

Three different types of techniques have been employed to describe the different chemerin forms present in biopsy samples: purification and characterization by mass spectrometry, activity assays, and immune-based methods. Neither SDS-PAGE nor ion-exchange chromatography has sufficient resolution to clearly separate the different chemerin forms and therefore more laborious methods have been used. Purification followed by mass spectrometry to determine precise mass is the gold standard, and has been used to identify new forms and to validate preparations of chemerin forms [79]. In a recent study, the method was refined to include the use of stable isotope-labeled standards, and was used to determine the chemerin forms present in human biofluids [80]. This study, however, failed to include methods to inhibit ex vivo proteolysis, and did not demonstrate the recovery of a “spiked” sample of the precursor (chem163S) or the fully active form, chem157S.

The relative activity of different chemerin forms varies from fully active forms such as chem157S, partially active forms such as chem158K, to completely inactive forms such as chem155A. Based on these differences in activity, assays have been used to distinguish between active and inactive chemerin forms. This technique has allowed the discrimination in serum between control and obese mice based on the different levels of activity [62]. Activity assays allow the assessment of the potential of the chemerin in a biopsy sample to have biological relevance via its ability to trigger chem1 or chem2 signaling. They have the disadvantage of not giving precise mass determinations of each chemerin form.

The majority of reports on chemerin levels in biopsy samples have employed commercial ELISAs for assaying them. ELISAs are fast and convenient with no special equipment required and can be run on large numbers of samples. However, there are problems with the commercial ELISAs. These include the lack of validation by their manufacturers on a mammalian-produced standard. Instead, the manufacturers have assumed that *E. coli*-produced material has an equivalent specific activity to mammalian material. In addition, there are no data available from the manufacturers on either the binding sites for the antibodies used in the ELISAs or the dose response curves for the different chemerin forms.

The behavior of the different chemerin forms in the ELISA is critical for interpreting results from biopsy samples. One commercial ELISA detects the different chemerin forms with parallel dose response curves, but the EC_50_ differs between the forms to such an extent that the assay is three-fold more sensitive for the active chem157S than the precursor chem163S (Figure 6) [79]. Consequently, a change in the levels of chemerin measured by this ELISA could be due to either a real change in the mass of chemerin or to a change in the proportion of the different forms in the samples. No other commercially available ELISA has been characterized in this detail.

To overcome these problems, ELISAs specific for many of the chemerin forms described in the literature have been developed for both human and mouse chemerin forms [36,38,79]. The specificity of those ELISAs relies on the specificity of the antibodies being used, and their accuracy depends on the quality of the standards. In these studies, the standards were produced by the expression of the different chemerin forms in CHO cells [27]. The polyclonal antibodies for the specific forms were raised in either rabbits or chickens. To develop a validated ELISA for a specific form requires using a peptide antigen based on the C-terminus of the specific form for which the ELISA is being developed, in order to raise the antibody which is then purified by chromatography on a column of the peptide antigen [25,36,79]. The immunoglobulin is subsequently purified on a column containing peptides representing the C-termini of the other chemerin forms, and the unbound fraction is then tested for specificity against purified chemerin form standards by Western and binding assays. In the case of some of the specific ELISAs, it was necessary to include peptides from other forms in the detection step in the ELISA when the sample is incubated with the immobilized antibody on the plate, in order to further improve specificity.

Unfortunately, neither those reagents nor the ELISAs are commercially available. Moreover, to fully profile a biopsy sample using ELISAs requires a one-step purification on a heparin affinity column, followed by a total of 7 ELISAs in either humans or mice—one ELISA to measure the level of all chemerin forms (total chemerin), and 6 specific ELISAs, one each for chem163S, chem158K, chem157S, chem156F, chem155A, and chem144D, a chemerin form identified from plasma from patients about to undergo bariatric surgery [79]. The fact that this method is very labor-intensive, coupled with the need for the specific antibodies, means that it is not generally available.

## 9. Chemerin Forms and Disease

Since chemerin was first identified, investigators have shown in longitudinal, cross-sectional, and case control studies, as well as a few prospective studies, that there are differences in chemerin levels between patients with disease and normal samples. Here a couple of indications will be discussed but more complete reviews of this topic are available [69,81,82,83,84]. These studies have been able to show a correlation between chemerin levels and disease incidence, severity, or progression. That correlation between chemerin and disease does not prove that chemerin is causing the disease as, alternatively, the change in chemerin levels could be caused by the disease, or an independent unknown factor could be causing both the disease and the changes in chemerin levels.

Single nucleotide polymorphisms (SNP) have been identified in the *RARRES2* gene but none of the ones used in studies investigating the association with disease so far has been an informative SNP that alters the encoded amino acid sequence. The association studies have in some cases shown a positive association with various phenotypes, for example with regional body fat disposition [85], gestational diabetes [86], or serum chemerin levels [87]. Other studies failed to find an association between *RARRES2* SNPs and type-2 diabetes (T2D) [88]. These differences found in the effect of SNPs within the *RARRES2* gene on diseases might be due to the different populations studied or the definitions of the indications.

There is a lack of information about the normal variation in chemerin levels in individuals, with the levels reported in research papers varying from median levels of <10 ng/mL to ones reporting median levels of >100 ng/mL [79,89]. The field needs to develop a method to normalize these data, such as creating an international standard for chemerin ELISAs and activity assays in order to improve the comparability of results from different laboratories. 

Chemerin levels did not change in post-prandial plasma samples compared with fasting samples in a small study which also showed no changes in the levels of the different forms [79]. In contrast, a drop in total chemerin levels was found following feeding with a larger reduction in people with obesity [89].

Diurnal variation in total serum chemerin levels has been observed in mice with differences in the diurnal rhythm found in wild type and obese mice [90], but no comparable studies have been reported for humans.

There are limited comparative data on any potential differences in circulating chemerin levels in men and women; some studies have shown that women have higher chemerin levels than men [91,92], but other studies have reached the opposite conclusion [93]. Although reports are available on chemerin levels in different racial or ethnic groups, the influence of this factor on circulating chemerin has not been formally compared within a single study to our knowledge. Information and an agreed consensus about these baseline parameters would improve the interpretation of studies investigating chemerin in disease. 

### 9.1. Cardiovascular Disease

A role for chemerin in hypertension has been shown in rats and humans [94], while atherogenesis is also affected by chemerin [40]. The mechanism by which chemerin in hypertension regulates blood pressure has been explored in rats, as mice do not show the same response to chemerin in aorta and arteries as humans, while rats do [51]. The chemerin is produced by the peri-vascular adipose tissue and causes contraction by acting on vascular smooth muscle cells via chem1. Despite the lack of effect reported on arterial contraction by the nonapeptide, mice given chemerin (6 μg/kg/day) intraperitoneally had elevated systolic blood pressure [95]. These mechanistic studies and model organism experiments show that chemerin affects blood pressure, but unfortunately association studies on human hypertension and the levels of different forms of chemerin are currently lacking. 

In a study on a large cohort derived from the European prospective investigation into nutrition and cancer (EPIC), chemerin levels in plasma were measured by ELISA. Participants in the quartile with the highest chemerin levels had a hazard ratio of 3.57 compared to the those in the lowest quartile for the development of myocardial infarction, stroke, and type 2 diabetes (T2D) [96]. Importantly, this was a prospective study in which the incident cases occurred after the acquisition of the samples. When the data were adjusted for body mass index (BMI) and waist circumference, the hazard ratio was not significantly changed. This study suggests that plasma chemerin levels could be a biomarker for cardiovascular risk.

A recent meta-analysis that included five independent studies confirmed that in patients with diagnosed acute coronary syndrome (ACS), chemerin levels were higher than in controls [97]. This was not found for patients with stable angina pectoris. In two studies that investigated T2D in patients with ACS, levels of chemerin were higher in T2D patients than those without. 

In patients with congestive heart failure (CHF), serum chemerin levels were measured while outcomes were investigated prospectively [98]. CHF patients whose chemerin levels were in the top quartile had more major adverse cardiac events than those with chemerin levels in the bottom quartile, with a hazard ratio of 3.25, while the hazard ratio for all cause mortality was 3.06. Analysis of Kaplan–Meier event-free survival curves showed that high chemerin levels were a predictor of major adverse cardiac events irrespective of the level of N-terminal prohormone of brain natriuretic peptide (BNPT), a well-documented predictive biomarker in CHF.

### 9.2. Chemerin in Obesity and Diabetes

Data from cell and animal studies have documented that chemerin is involved in energy balance and metabolism with implications for obesity, diabetes, and metabolic syndrome in adults [50,53,54,99]. The role of chemerin in pediatric and gestational diabetes will not be discussed here, although there is good evidence of its involvement [100,101]. There are many reports describing chemerin levels along with other adipokines in various cohorts showing that chemerin levels are associated with disorders involved with energy balance [81,82,92,102]. Those reports are in agreement that levels of circulating chemerin are higher in people with obesity.

In patients about to undergo bariatric surgery, there are higher levels of total chemerin in their circulation, with data suggesting that those levels are reduced at 6 or 12 months after surgery [103,104,105,106,107]. In contrast, in the days immediately after the surgery, there are inconsistent data on chemerin levels.

In one report, the levels of chemerin isoforms in plasma were also investigated. In a cohort that was 80% male, total chemerin was increased in plasma from patients with obesity compared to lean controls. The use of ELISAs specific for different chemerin forms showed that the level of the precursor, chem163S, was reduced and that the increase was due to chemerin degraded to a smaller size than chem155A [79]. One form of that degraded chemerin was identified as chem144D. In a study investigating a small number of female patients with obesity, the bioactivity of serum samples was measured [61]. Compared to normal BMI controls, patients with obesity had higher levels of total chemerin, which was more bioactive than the chemerin from controls. Both of these studies show that in patients with obesity, circulating chemerin levels are increased, with more processing occurring.

These results are similar to those obtained in a model of obesity in mice in which plasma chemerin forms were determined by specific ELISAs [38]. Those results were confirmed in a study in which the bioactivity of chemerin samples was assayed in serum from murine models of obesity [62].

In rodent animal models, the status of the chemerin system affects energy metabolism and the development of insulin resistance [108]. In rodents, loss- or gain-of-function mutations in the chemerin gene or either of its two receptors leads to changes in adipogenesis, food intake, glucose homeostasis, and inflammation both in control animals and in animals challenged with interventions such as a high fat diet [51,53,109,110]. These data support the hypothesis that chemerin is involved in the development of T2D.

That concept has been investigated in human clinical samples, and in numerous studies an association has been found between chemerin levels and diabetic and obesity-related comorbidities (summarized in [111]). For example, serum levels of total chemerin have been found to be positively associated with levels of hemoglobin A1c (Hb A1c) in a large population of relatively healthy Taiwanese [92]. In this study, a small difference was detected in chemerin levels between men and women. Multivariate analysis showed that several markers of metabolic disease, such as blood pressure, cholesterol, and triglyceride levels, were correlated positively with serum chemerin levels, while HDL levels were negatively correlated with chemerin levels. Both BMI and Hb A1c were independent predictors of chemerin levels, suggesting that obesity and glucose homeostasis were affecting chemerin levels by different mechanisms.

Some studies have presented data that are not in agreement with the idea that T2D and metabolic syndrome patients have increased chemerin levels. For example, in a study in which Japanese with T2D or metabolic syndrome were compared to controls, men but not women were found to have lower chemerin levels than either people with metabolic syndrome or controls [93]. The disagreements between studies could arise from differences in which factors, such as age or sex, if any, were taken into account in the statistical analysis.

## 10. Discussion and Future Directions

Chemerin is a small molecule with pleiotropic functions that acts both through its receptors and as an anti-bacterial defensin. It has been shown to play a role in a variety of inflammatory and metabolic conditions as well as in cancer [83]. In animal models, genetic modifications to increase or reduce expression of the components of the chemerin system have shown a physiological or pathophysiological role for chemerin in some conditions such as obesity, diabetes, myocardial infarction, and stroke.

In humans, those studies have shown associations between circulating chemerin levels and various conditions, but definitive mechanistic links have not yet been established. The association studies that have established the correlation between chemerin levels and various disease conditions would be enhanced by including analysis of the *RARRES 2* gene SNPs alongside the chemerin measurements.

Since its original discovery, it has been appreciated that chemerin was produced as a precursor that needed to be activated before it could cause its receptors to signal. Despite this insight, the analytic methods to characterize chemerin forms in biopsy samples from animal models and humans have not been readily available. Simple, cheap, and readily accessible methods for assaying the different forms would allow researchers to describe their samples more precisely. Non-expert readers of papers on chemerin would benefit from the inclusion of caveats discussing the limitations of the quantitation methods employed, such as activity assays and ELISAs.

The reported median levels of circulating chemerin differ by more than an order of magnitude. A consensus on normalizing the values from studies carried out in different labs using different methods would be very helpful in promoting the potential of developing chemerin determinations as a disease biomarker.

Chemerin is an exciting molecule, with possibilities of intervening in the chemerin system to treat or prevent various disease conditions. To carry out the studies required for this, better tools need to be generated, such as a specific and potent antagonist of chem2 as well as improving the availability of the specific antagonist of chem1, CCX832, to researchers [51]. There have been no agonists described to our knowledge that are specific for either chem1 or chem2. The interpretation of experiments using chemerin proteins, the nonapeptide, or peptidomimetics as pharmacological tools would be easier if their distribution, clearance, and pharmacodynamics had been described.

## Figures and Tables

**Figure 1 biomedicines-10-02018-f001:**
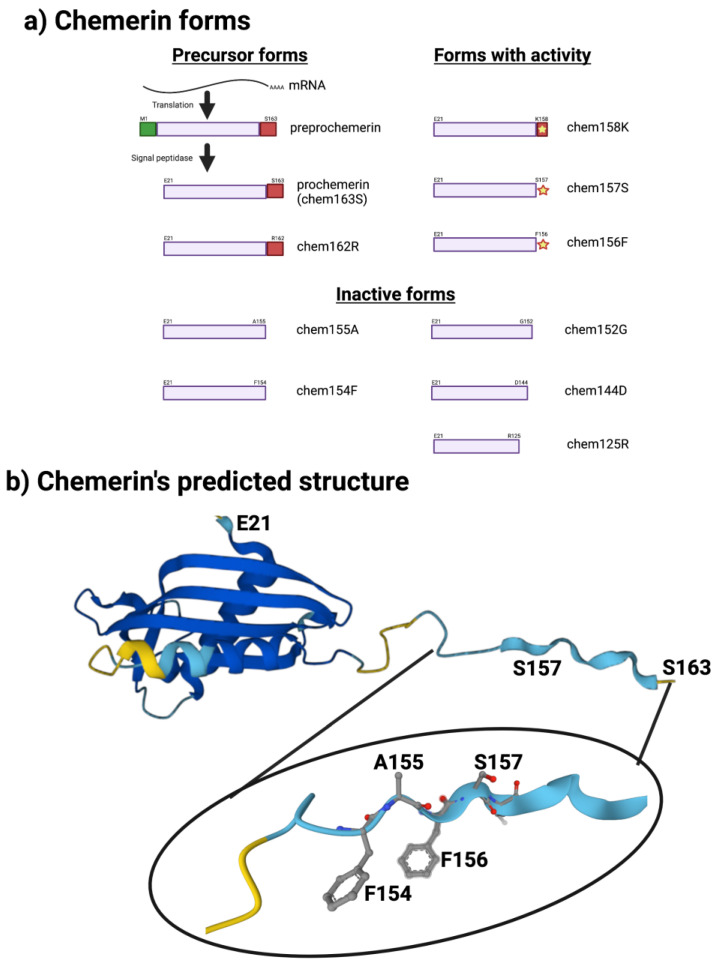
Chemerin forms and chemerin’s predicted structure. Created with BioRender.com. (**a**) Chemerin forms that have been identified in biopsy samples are grouped into those that are precursors with no intrinsic activity but can be activated by appropriate proteolysis, those that are active, and those are inactive. (**b**) The structure of chemerin predicted by AlphaFold [15,16] with the darker blue colors representing higher per-residue confidence in the prediction with a magnification of the C-terminal tail.

**Figure 2 biomedicines-10-02018-f002:**
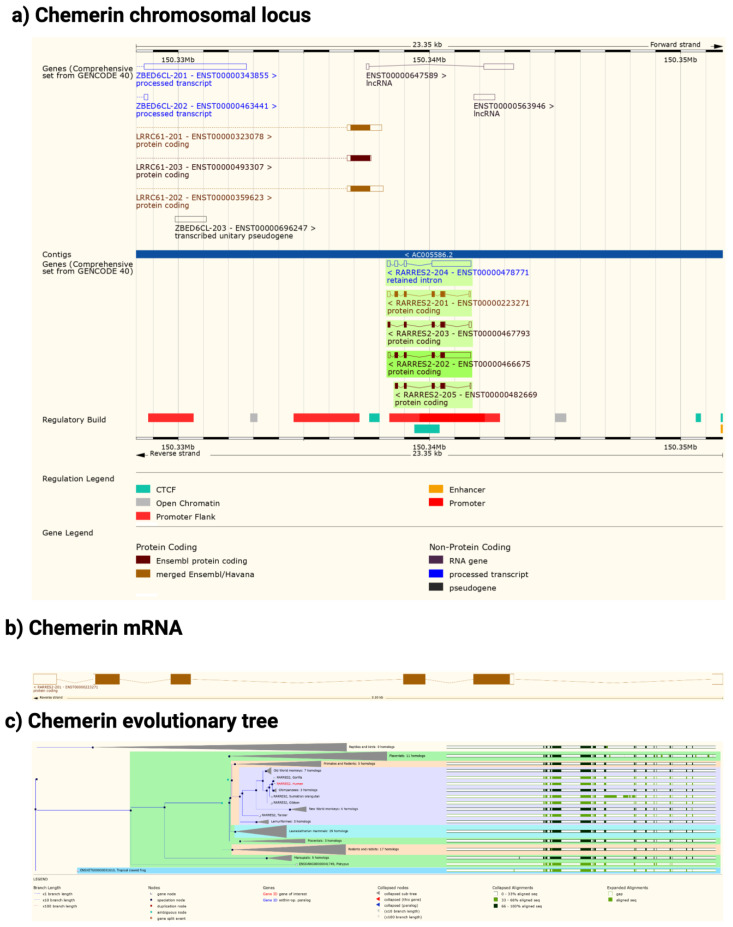
Chemerin gene (*Rarres2*), mRNA, and evolutionary tree. (**a**) Chromosomal neighborhood of the chemerin gene on chromosome 7. Chemerin in green on anti-sense strand. (**b**) Chemerin mRNA with 6 exons in boxes, filled if coding. (**c**) Evolutionary tree for the chemerin gene. Figures created from data in EMSEMBL human genome build GRCH38.p13 (https://ensembl.org/Homo_sapiens accessed on 11 August 2022).

**Figure 3 biomedicines-10-02018-f003:**
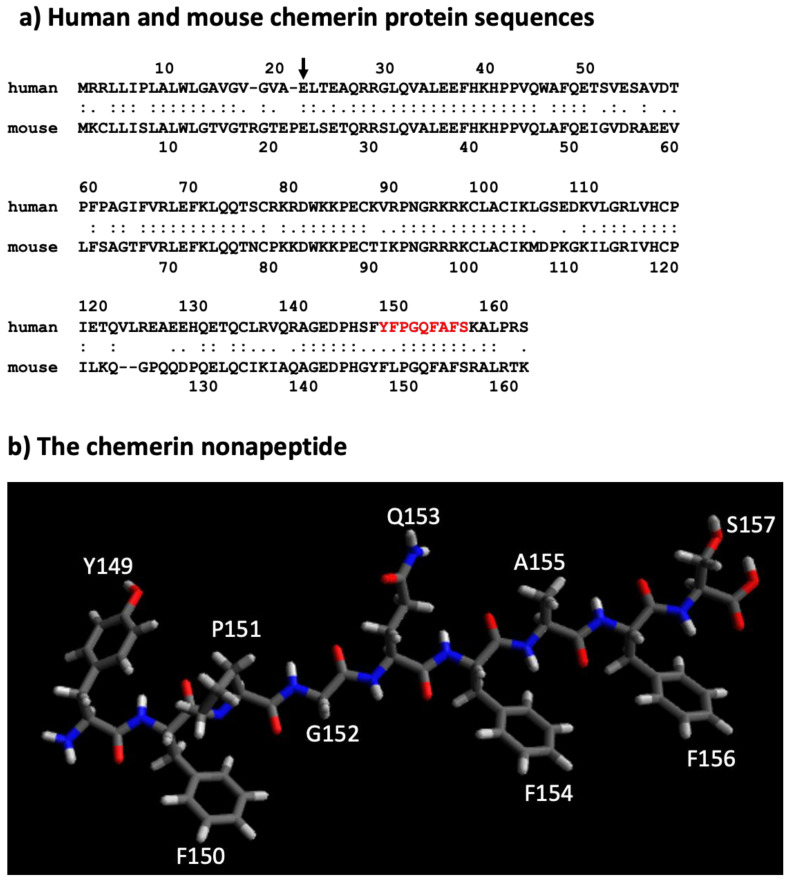
Sequences of mouse and human chemerin and structure of nonapeptide. (**a**) Mouse and human chemerin protein sequences. The arrow marks the site of cleavage by the signal peptidase, and the amino acid residues highlighted in red represent the nonapeptide. (**b**) chemerin nonapeptide.

**Figure 4 biomedicines-10-02018-f004:**
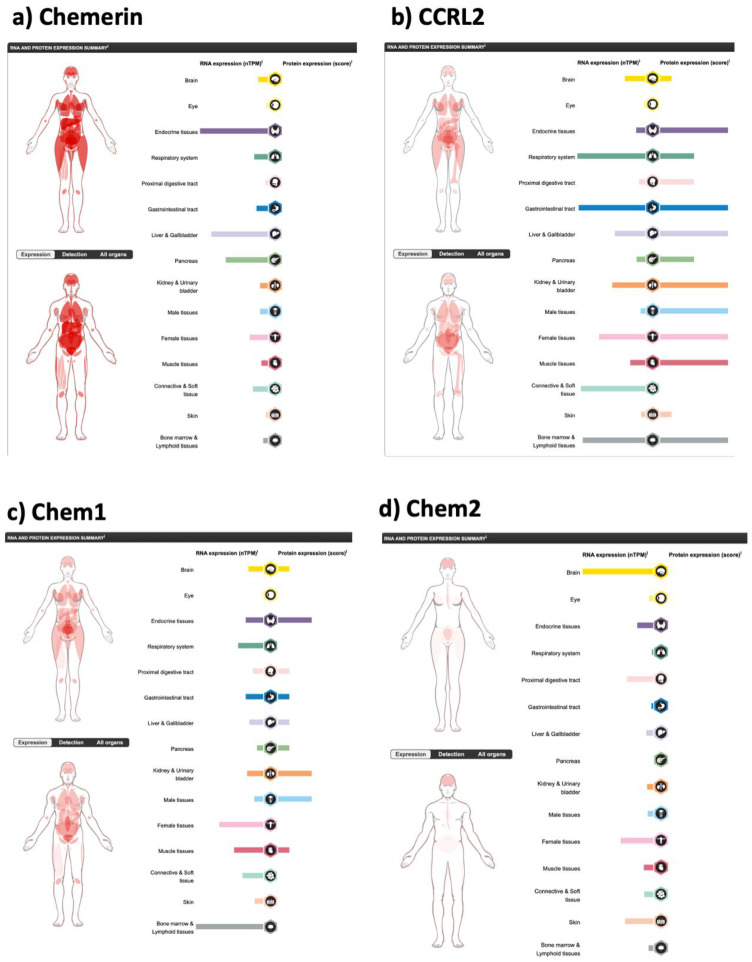
Expression of the components of the chemerin system in different tissues, on the left showing the mRNA expression data for each in normalized transcripts per million (nTPM), and on the right, where available, the score is the protein expression data based on immunohistochemical data manually scored with regard to staining intensity (negative, weak, moderate or strong) and fraction of stained cells (<25%, 25–75% or >75%). Each combination of intensity and fractions is automatically converted into a protein expression level score as follows: negative—not detected; weak <25%—not detected; weak combined with either 25–75% or 75%—low; moderate <25%—low; moderate combined with either 25–75% or 75%—medium; strong <25%—medium, strong combined with either 25–75% or 75%—high. In addition to this, protein expression values are manually adjusted as necessary when evaluated by our expert annotators. (**a**) chemerin (*RARRES2*), (**b**) CCRL2, (**c**) chem1 (CMKLR1) and (**d**) chem2 (GPR1). Data from the Human Protein Atlas v21.1 [60] (https://www.proteinatlas.org/ENSG00000106538-RARRES2/tissue, https://www.proteinatlas.org/ENSG00000174600-CMKLR1/tissue, https://www.proteinatlas.org/ENSG00000121797-CCRL2/tissue and https://www.proteinatlas.org/ENSG00000183671-GPR1/tissue accessed on 11 August 2022).

**Figure 5 biomedicines-10-02018-f005:**
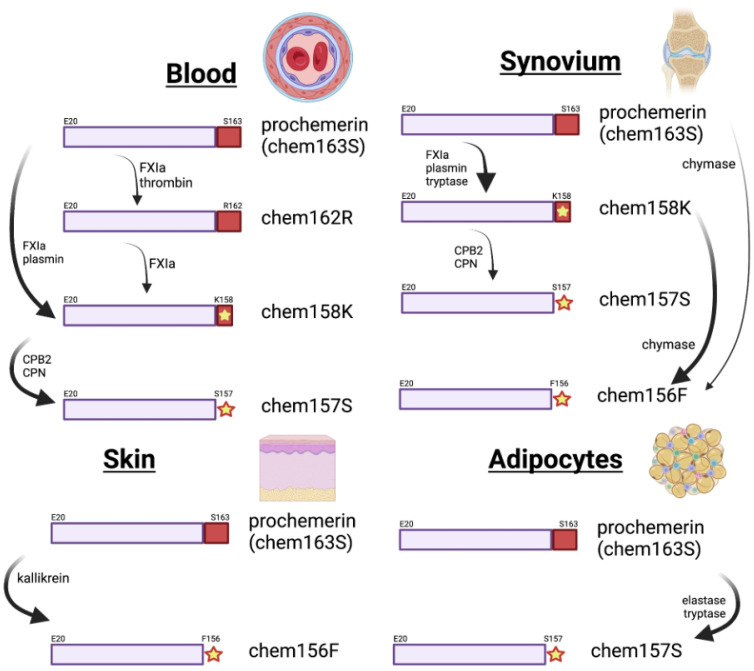
Processing of chemerin in various tissues or cells. These diagrams depict the enzymes that have been proven to be involved in chemerin processing in the tissues or cells. In blood, the cleavages are shown following activation of coagulation and fibrinolysis; in synovium, from patients with rheumatoid arthritis; in skin, in patients with psoriasis, and in adipocytes that have been stimulated with TNF-α. Created with BioRender.com.

**Figure 6 biomedicines-10-02018-f006:**
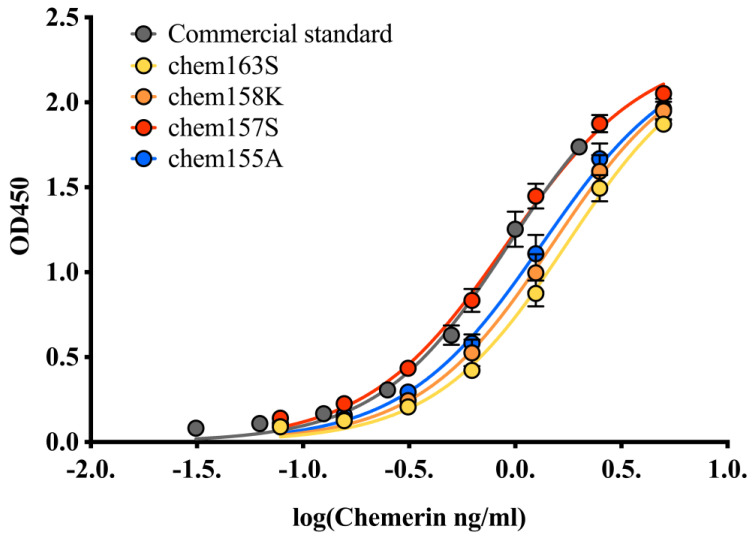
Comparison of the potency of different chemerin forms produced in a mammalian expression system [27] in a commercial ELISA. Standard curves in that commercial ELISA with the *E. coli*-derived, Glu^21^–Ser^157^ chemerin standard provided in the ELISA kit (gray), recombinant chem163S (yellow), recombinant chem158K (orange), recombinant chem157S (red), and recombinant chem155A (blue). Protein concentration for each standard protein was measured by Bradford assay prior to ELISA.

## Data Availability

Not applicable.

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
