# Peer review of "Chemerin Forms: Their Generation and Activity"

_biomedicines, 2022, doi:10.3390/biomedicines10082018_

Round 1
Reviewer 1 Report
The current manuscript authored by Lei Zhao et al. described “Chemerin forms: their generation and activity”. The authors discusses the specific tissue expression of the components of the chemerin system and the role of different proteases in regulating the activation and inactivation of chemerin. Although authors present a well review work there are number of issues which needs to be addressed:
1. Please change the page 1, lane 7 “RARRES 2” to “RARRES2”.
2. Please change the page 4, lane 81 “thaat” to “that”.
3. Please change the page 6, lane 115 “Rarrea2” to “RARRES2”.
4. Please delete the page 9, lane 244 “(?)”.
5. Please rewrite the Figure 4 ligand and the part of manuscript form lane 331 to 336.
6. Figure 4 need to revise.
7. Please change the page 14, lane 431 “In vitro” to “In vitro”.
Author Response
We appreciate the careful reading of our manuscript and have corrected the errors pointed out.
- Please change the page 1, lane 7 “RARRES 2” to “RARRES2”.
Done
- Please change the page 4, lane 81 “thaat” to “that”.
Done
- Please change the page 6, lane 115 “Rarrea2” to “RARRES2”.
Done
- Please delete the page 9, lane 244 “(?)”.
Done
- Please rewrite the Figure 4 ligand and the part of manuscript form lane 331 to 336.
Figure 4 legend has been revised to include more details about the information shown..
- Figure 4 need to revise.
Fig 4 has been revised so that the upper labels are visible.
- Please change the page 14, lane 431 “In vitro” to “In vitro”.
Done
Reviewer 2 Report
This review concerning generation and activity of chemerin forms is very interesting and instructive. The authors have performed a through analysis of the literature published on this topic during the last twenty years and have offered a clear and synthetic account – easy to follow – on the chemerin system of proteins and receptors, on chemerin forms and on the involvement of chemerins in cardiovascular and metabolic pathology, suggesting also the possible use of chemerins as biomarkers.
Besides all positive aspects, a few small points might be addressed:
1. Abstract, line 17 – some examples of pathology – as a preview – might be given; same thing in discussions (line 643).
2. Section 5 – the same idea concerning chem157S is repeating in this section (lines 156-160 / 215-218); one of them might be reformulated, only if possible. In line 244 seems that something is missing; a reference maybe?
3. Section 7 – it seems (line 342) that the text is referring to figure 5, not 6.
4. Section 9 – is very instructive concerning the relation between chemerins and pathology. The relation with diabetes and obesity is very developed. Is it possible to develop a little more the relation with the cardiovascular diseases? Are there any meta-analyses comparing the effect of hypertension on chemerin levels?
Author Response
We thank the reviewer for their comments and have responded as described below in italics to the specific points.
- On page 4 (lines 87-88 and 106-109), the authors state that chemerin is not present in fish, which leads to suggestions regarding the role of other ligands such as resolvins as ligands of the chem receptors in these species. However, although chemerin genes are not annotated as such for fish species in the Ensembl database, they are present in all of them. The structure and function of the chemerin system in zebrafish and other fish species will be published pretty soon. It might be more relevant to state that “chemerin genes have not been described so far in fish” and avoid driving conclusions from their absence, as this will rapidly become inexact.
We thank the reviewer for the zebra fish information and agree with them that the statements about chemerin being missing in bony fish overstated the available data. That point has been revised to describe the situation and now states that there is no annotated chemerin gene found in bony fish in the NCBI or ENSEMBL databases. The statement on alternative ligands has also been rewritten.
- On page 7 (lines 174-178), the authors state that the chemerin nonapeptide has an activity almost equivalent to that of full size chemerin (3-fold change), concluding that most of the binding energy is located in the C-terminal tail. Actually, the EC50 of the reference chemerin used by most investigators (R&D Systems) is about 0.2 nM, while the EC50 of the peptide is 5-10 nM in most studies. This is more like a 50-fold difference, suggesting a significant contribution of the cystatin-fold domain in the binding energy. It remains true that the C-terminal domain is essential for activating the receptor.
The reviewer is correct that a range of activities have been reported for chem157S and the nonapeptide on chem1. There are, however, only 3 determinations of EC50 that we were able to find in the literature (see Table 1 below) plus the report from de Henau et al. (2016) which determined the signaling parameters for 5 Ga proteins plus 2 b-arrestins. The Table shows EC50 for the mammalian expressed chem157S was low nM while the E. coli expressed chem157S was 21 nM with no error stated. The EC50 values reported by de Henau et al (2016) for BRET assays for b-Arrestins were also higher than for those for Ga proteins.
We were unable to find a determination of the EC50 for the R&D chemerin although Li et al. 2014 reported a binding constant of 0.3 nM. Toulany et al. (2016) showed a dose response curve for R&D chem157S in a b-arrestin2 assay with values in single-digit pM (Figure 1) but the accompanying table is given in nM (Table 2) and no EC50 is presented.
The nonapeptide gave EC50 values ranging from low nM up to low 100s nM (Table 1) depending on the assay employed. The low values were in assays in which all Ga proteins participate while the higher values were in BRET assays on individual Ga proteins detecting their conformational change in response to ligand binding.
Table 1: chem157S and the nonapeptide activity on chem1
|
|
|
Chemerin source |
assay |
Protein EC50 (nM) |
Nonapeptide EC50 (nM) |
|
Wittamer et al |
2004 |
CHO cells |
Aequorin |
4.55 ± 1.07 |
3.46 ± 1.13 |
|
Yamaguchi et al |
2011 |
CHO cells |
Ca++ flux |
1.17 ± 0.74 |
5.92 ± 3.26 |
|
Yamaguchi et al |
2011 |
CHO cells |
chemotaxis |
3.15 ± 1.57 |
6.38 ± 3.03 |
|
Schultz et al |
2013 |
E. coli |
Inositol phosphate |
21 |
3.5 |
|
De Henau et al |
2016 |
CHO cells |
Gai1/BRET |
2.82 ± 1.62 |
141 ± 15.5 |
|
De Henau et al |
2016 |
CHO cells |
Gai2/BRET |
2.82 ± 2.0 |
141 ± 20.0 |
|
De Henau et al |
2016 |
CHO cells |
Gai3/BRET |
8.13 ± 1.38 |
240 ± 18.6 |
|
De Henau et al |
2016 |
CHO cells |
Gaoa/BRET |
5.25 ± 1.62 |
120 ± 18.6 |
|
De Henau et al |
2016 |
CHO cells |
Ga0b/BRET |
7.59 ± 2.04 |
107 ± 14.1 |
|
De Henau et al |
2016 |
CHO cells |
b-Arr1/BRET |
32.4 ± 12.9 |
ND |
|
De Henau et al |
2016 |
CHO cells |
b-Arr2/BRET |
10 ± 1.23 |
ND |
In unpublished work, we tried to produce various forms of chemerin in E. coli using various different expression systems to form cytoplasmic chemerin, in inclusion bodies and secreted. As reported by Schultz et al. (2014), chemerin readily forms inclusion bodies that can be solubilized. When analyzed by non-reduced SDS-PAGE or SEC, large multimers were present in the preparations irrespective of the solubilization conditions or if redox systems were included. This resulted in activity determinations being compromised by these aggregates. It is important to note that all of the activity assays (except for the anti-bacterial assays) depend only on the nonapeptide and so the overall integrity of chemerin is not being measured. Analysis of R&D’s chem157S by non-reducing SDS-PAGE or SEC is not available in their product literature.
The EC50 previously reported by us (Yamaguchi et al. 2011) for chem157S was based on 5 independent preparations of chem157S, each of which was measured on 3 independent occasions with triplicates for each point on the dose response curve (unpublished data). Our EC50 was similar to that reported by Wittamer et al (2004) and de Henau et al (2016) who also produced chem157S from CHO cells.
The sequence of amino acids in chemerin between amino acids #148-156 (FYFPGQFAF) is hydrophobic and peptides containing them may present solubility issues in aqueous solutions. This is the case with the nonapeptide which we found needed to be solubilized in 10% DMSO before dilution into assay buffer (unpublished data). Failure to dissolve the peptides into a mono-molecular solution leads to an EC50 determination that is higher than with a validated mono-molecular solution.
Based on this discussion, we think that the EC50 values given by Wittamer et al. (2004) and Yamaguchi et al. (2011) are the best available estimates for both chem157S and the nonapeptide. The values for chem157S have been confirmed by de Henau et al (2016) in their studies on signaling parameters of chemerin. These authors conclude that the difference in EC50 between chem157S and the nonapeptide is ~3 fold as we stated on line 189.
A new paragraph has been added starting at line 202 that explains for the reader these issues and the text has been modified elsewhere in this section to clarify these points.
- There are a number of misspellings: “thaat” on line 81, “represnting” on line 84, “mRNAshave” on line 101, “assay(?)” on line 244, “in cleage by” on line 342…
Corrected
Reviewer 3 Report
The manuscript by Zhao et al. reviews the literature regarding the chemerin system, focusing on the different forms of the ligands, the way they are generated and the correlation with disease states.
Altogether, the manuscript is well written and represent a faithful reflect of the currently available data. I have only a small number of comments and suggestions.
1. On page 4 (lines 87-88 and 106-109), the authors state that chemerin is not present in fish, which leads to suggestions regarding the role of other ligands such as resolvins as ligands of the chem receptors in these species. However, although chemerin genes are not annotated as such for fish species in the Ensembl database, they are present in all of them. The structure and function of the chemerin system in zebrafish and other fish species will be published pretty soon. It might be more relevant to state that “chemerin genes have not been described so far in fish” and avoid driving conclusions from their absence, as this will rapidly become inexact.
2. On page 7 (lines 174-178), the authors state that the chemerin nonapeptide has an activity almost equivalent to that of full size chemerin (3-fold change), concluding that most of the binding energy is located in the C-terminal tail. Actually, the EC50 of the reference chemerin used by most investigators (R&D Systems) is about 0.2 nM, while the EC50 of the peptide is 5-10 nM in most studies. This is more like a 50-fold difference, suggesting a significant contribution of the cystatin-fold domain in the binding energy. It remains true that the C-terminal domain is essential for activating the receptor.
3. There are a number of misspellings: “thaat” on line 81, “represnting” on line 84, “mRNAshave” on line 101, “assay(?)” on line 244, “in cleage by” on line 342…
Author Response
We thank the reviewer for their comments and have responded below in italics to the specific points..
Besides all positive aspects, a few small points might be addressed:
- Abstract, line 17 – some examples of pathology – as a preview – might be given; same thing in discussions (line 643).
Based on this suggestion examples have been added in the abstract and to the discussion.
- Section 5 – the same idea concerning chem157S is repeating in this section (lines 156-160 / 215-218); one of them might be reformulated, only if possible. In line 244 seems that something is missing; a reference maybe?
Thank you bringing the repetition to our attention. We have removed sentences from lines 158-160 (previous numbering) and added some more information to the paragraph beginning line 231. References have been added to line 244 (old numbering).
- Section 7 – it seems (line 342) that the text is referring to figure 5, not 6.
Corrected
- Section 9 – is very instructive concerning the relation between chemerins and pathology. The relation with diabetes and obesity is very developed. Is it possible to develop a little more the relation with the cardiovascular diseases? Are there any meta-analyses comparing the effect of hypertension on chemerin levels?
A new paragraph has been added to discuss chemerin and hypertension to this section. A PubMed search on hypertension and chemerin returned no clinical studies at all.
Round 2
Reviewer 1 Report
All comments were done by the authors.